# Exemplars in vaccine delivery protocol: a case-study-based identification and evaluation of critical factors in achieving high and sustained childhood immunisation coverage in selected low-income and lower-middle-income countries

Robert A Bednarczyk [iD],[1,2] Kyra A Hester,[3] Sameer M Dixit,[4] Anna S Ellis,[3] Cam Escoffery,[5] William Kilembe,[6] Katie Micek,[3] Zoë M Sakas,[3] Moussa Sarr [iD],[7] Matthew C Freeman[3]

For numbered affiliations see end of article.

**Correspondence to**
Dr Robert A Bednarczyk; rbednar@emory.edu

## ABSTRACT

**Introduction** Increases in global childhood vaccine delivery have led to decreases in morbidity from vaccine-preventable diseases. However, these improvements in vaccination have been heterogeneous, with some countries demonstrating greater levels of change and sustainability. Understanding what these high-performing countries have done differently and how their decision-making processes will support targeted improvements in childhood vaccine delivery.

**Methods and analysis** We studied three countries— Nepal, Senegal, Zambia—with exemplary improvements in coverage between 2000 and 2018 as part of the Exemplars in Global Health Programme. We apply established implementation science frameworks to understand the 'how' and 'why' underlying improvements in vaccine delivery and coverage. Through mixed-methods research, we will identify drivers of catalytic change in vaccine coverage and the decision-making process supporting these interventions and activities. Methods include quantitative analysis of available datasets and in-depth interviews and focus groups with key stakeholders in the global, national and subnational government and non-governmental organisation space, as well as community members and local health delivery system personnel.

**Ethics and dissemination** Working as a multinational and multidisciplinary team, and under oversight from all partner and national-level (where applicable) institutional review boards, we collect data from participants who provided informed consent. Findings are disseminated through a variety of forms, including peer-reviewed manuscripts related to country-specific case studies and vaccine system domain-specific analyses, presentations to key stakeholders in the global vaccine delivery space and narrative dissemination on the Exemplars.Health website.

### Strengths and limitations of this study

► This study is led by a multidisciplinary team and grounded in several theoretical frameworks across disciplines from implementation science to behavioural theory.

► We used a cross-cutting, cross-disciplinary, approach, which assessed relevant domains across our selected exemplars countries as well as within the subjects that arise from the data, over a roughly 20-year time horizon.

► We selected three countries with historically high unvaccinated populations to represent different geographies, cultures and governments, as well as to highlight regions with historically high unvaccinated populations.

► We did not study a less successful, or 'non-exemplar', counterfactual country.

► The research tools identified and explored catalytic events and the implementation of external policies and development of internal policies and systems, with a focus on participants' current experiences and perceptions of prior activities.

## INTRODUCTION

Early childhood vaccination is widely recognised as one of the most important public health interventions. Increasing vaccine coverage globally has substantially reduced the incidence of, and mortality from, vaccine-preventable diseases.[1] While early childhood vaccine coverage has increased globally, there are still millions of children, particularly in low-income and lower-middle-income countries (LICs and LMICs,

respectively), who remain unvaccinated.[2] The WHO's Global Vaccine Action Plan sets global targets for all countries to achieve 90% national level coverage of diphtheria, tetanus, pertussis (DTP) for three doses of vaccine (DTP3), and 80% subnational level DTP3 coverage in every district by 2015.[3 4] Although significant progress has been made towards these goals—global DTP3 coverage increased from 72% in 2000 to 86% in 2018—the WHO/Unicef Estimates of National Immunisation Coverage (WUENIC) demonstrate that this progress fell short in both coverage and equity.[5] The COVID-19 pandemic has also negatively impacted routine immunisation globally; the extent of this impact is still being assessed,[6–8] and is outside of the scope of this retrospective evaluation.

The literature documenting identified barriers and facilitators of improved vaccine coverage is vast. The systematic review performed by Phillips *et al* provides a conceptual framework identifying facility readiness, intent to vaccinate and community access as the core determinants of effective vaccine coverage.[9] Similarly, LaFond *et al* identified direct and enabling drivers of immunisation coverage improvement as well as essential health and immunisation system components, such as district management teams and existence of basic routine immunisation resources and capacity.[10]

Identification of these barriers and facilitators is only a first step towards improving global vaccine coverage. There remains an evidence gap in understanding 'how' and 'why' these factors influence system performance. Notably, to strengthen immunisation programme function we need to understand the development, implementation and adaptation of programmes and interventions. Little rigorous evidence is available on the specific paths to success, including implementation strategies, in the LICs and LMICs that have achieved high and sustained immunisation coverage.

We apply a 'positive deviant' approach to study high-performing countries, that is, to understand successful vaccine system performance by identifying positive outliers—countries or systems that exceed their peers—and studying the factors that supported catalytic growth to reach a high level of coverage.[11] Through the identification of the components and pathways to high vaccine coverage among exemplar countries, actionable recommendations can be developed and disseminated to other countries that have not yet had similar success. These recommendations can support decision-making processes to improve immunisation programmes and health systems, improve overall vaccine coverage, and mitigate inequities in subnational vaccine coverage in these countries.

The Exemplars in Vaccine Delivery—nested within the larger Exemplars in Global Health partnership, aims to identify the 'how' and 'why' behind implementation of particular systems and decisions that led to high and sustained infant vaccine coverage through a geographically diverse set of positive deviant countries (ie, Nepal, Senegal, Zambia).[12] Using two complementary

implementation science frameworks and a multidisciplinary approach—reaching beyond medical and public health research—we built on the existing evidence and frameworks to explore specific components or critical factors of the immunisation system to identify potential areas of future research and investment in immunisation system interventions. This manuscript presents our mixed-methods data collection methods to address these outstanding questions.

## METHODS AND ANALYSIS
### Overview
The purpose of this study is to assess 'how' and 'why' some countries have succeeded in achieving significantly improved coverage rates between 2000 and 2018, and to provide actionable recommendations for improving national and subnational vaccine coverage. This study focuses on critical policy and programmatic innovations that drove changes to vaccine coverage and equity across the three countries of interest, and specifically investigates 'how' and 'why' these innovations were implemented.

Our research consortium includes Emory University, Georgia Institute of Technology, the University of Delaware, the Center for Molecular Dynamics in Nepal, the Center for Family Health Research in Zambia, the Institut de Recherche en Santé de Surveillance Epidemiologique et de Formation (Institute for Health Research, Epidemiological Surveillance and Training) in Senegal.

### Selection of exemplar countries
Three exemplar countries—Nepal, Zambia and Senegal—were selected based on available data and expert review. Countries were eligible for inclusion if, in the year 2000, (1) their population exceeded 5 million and (2) the World Bank classified them as low income. Forty-seven countries met these criteria. Two analyses were performed to identify exemplars from the eligible countries based on measured coverage of DTP1 and DTP3: direct estimates of the compound annual growth rate (CAGR) of vaccine coverage and a segmentation analysis based on coverage, dropout rates and country conflict status (figure 1). Taken together, DTP1 and DTP3 serve as common proxies for the function of the vaccine delivery system in each country, as DTP1 can indicate how many children are reached by the immunisation system, and DTP3 can indicate how many children the programme has continued to reach.[13 14]

The CAGR analysis used both WUNEIC and Institute of Health Metrics and Evaluation (IHME) data.[5 15] For the above-mentioned 47 countries, we calculated CAGRs for each country, with both WUENIC and IHME data, from 2000 to 2016. CAGR calculations used 3-year rolling averages. We found the highest-performing countries by applying predetermined cutoffs by data source; the cutoff percentage depended on the overall performance of the group. The WUENIC data had a CAGR cut-off of 0.9%, indicating a 9% increase over 10 years, and the

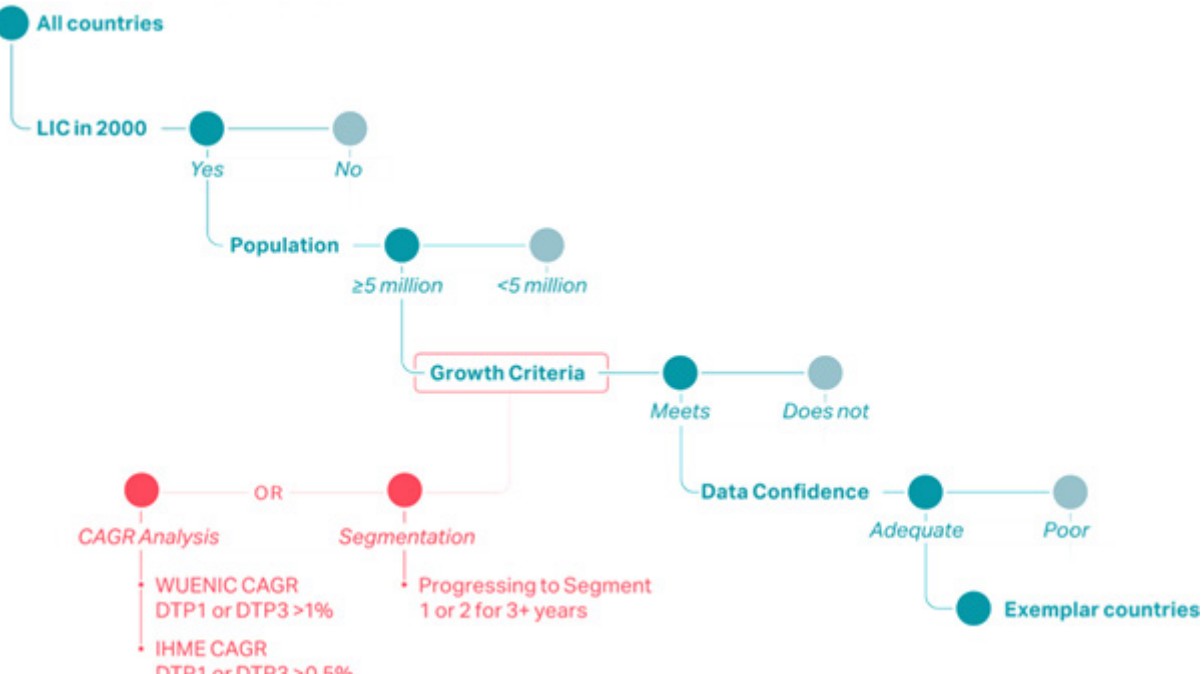

**Figure 1** Country filtering process, of which 47 countries met the growth criteria. CAGR, compound annual growth rate; DTP, diphtheria, tetanus, pertussis; IHME, Institute of Health Metrics and Evaluation; LIC, low-income countries.

IHME data had a CAGR cut-off of 0.5%, indicating a 5% increase over 10 years. Seventeen countries met both the WUENIC and IHME CAGR cut-off percentage.

The segmentation analysis used the rolling 3 year averages obtained from WUENIC data.[10] Five segments were created by analysing and ranking DTP1 coverage, DTP3 coverage, dropout rates and conflict. The segments were classified as follows: Segment 1 countries had 'proven themselves' with national DTP3 coverage greater than 90%; Segment 2 included countries that were 'on the right track' with national coverages of DTP3 less than or equal to 90%, but DTP1 greater than 80% and a dropout rate greater than 10%; Segment 3 included countries that were 'getting children back into the system,' with national coverages of DTP3 90%, DTP1 80% and a dropout rate 10%; Segment 4 included countries that were still 'building essentials', with national coverages of DTP3 90%, DTP1 80% and no conflict at time of selection; and Segment 5 included countries with ongoing conflict at time of selection. Exemplar countries were identified as those meeting all three of the following criteria: (1) The country was in segments 3, 4 or 5 at any time during the period 2005–2010; (2) The country progressed to either segment 1 or 2; and (3) The country stayed in segment 1 or 2 for at least 3 years (figure 2).

The shortlist of possible exemplar countries, based on both analyses, had 13 countries (table 1). The final three countries were selected to represent geographical diversity (South Asia, East Africa, West Africa), as these regions have the majority of unvaccinated children globally. The democracy index, as defined by the 2018 Democracy Index, was used for framing the country selection and for exclusion criteria .[16] Final exemplar countries were selected in conjunction with our technical advisory group (TAG).

### Country-level data collection

We conducted research at different levels of the healthcare system for each country: the national level, three subnational regions/provinces and three districts per region/province for a total of nine districts. Our predetermined subnational region selection criteria differed by country, but one region in each country contained the capital city of the country, with the other two regions stratified on factors determined with input from the local study team (eg, high/low subnational immunisation coverage, rural/urban, road access/lack of road access, ethnic/religious minority/majority). Changes in subnational immunisation coverage over time were assessed using district-level data (figure 3A–C). Districts were selected based on country specific CAGR and DTP3 percentile cutoffs.

### In-country stakeholder identification

Alongside our network of in-country and regional collaborators and networks, we identify a comprehensive list of key stakeholders to include in data collection. We aim to identify both individuals who were in the related positions at the time of data collection, and those who previously held such positions to assess how programmatic changes were implemented and adapted over time. The generalised list of positions is documented in table 2; due to local context and health system structure, specific positions may differ by country. Specific categories and titles, and the number of related data collection activities, will be presented alongside country-specific analyses.

 

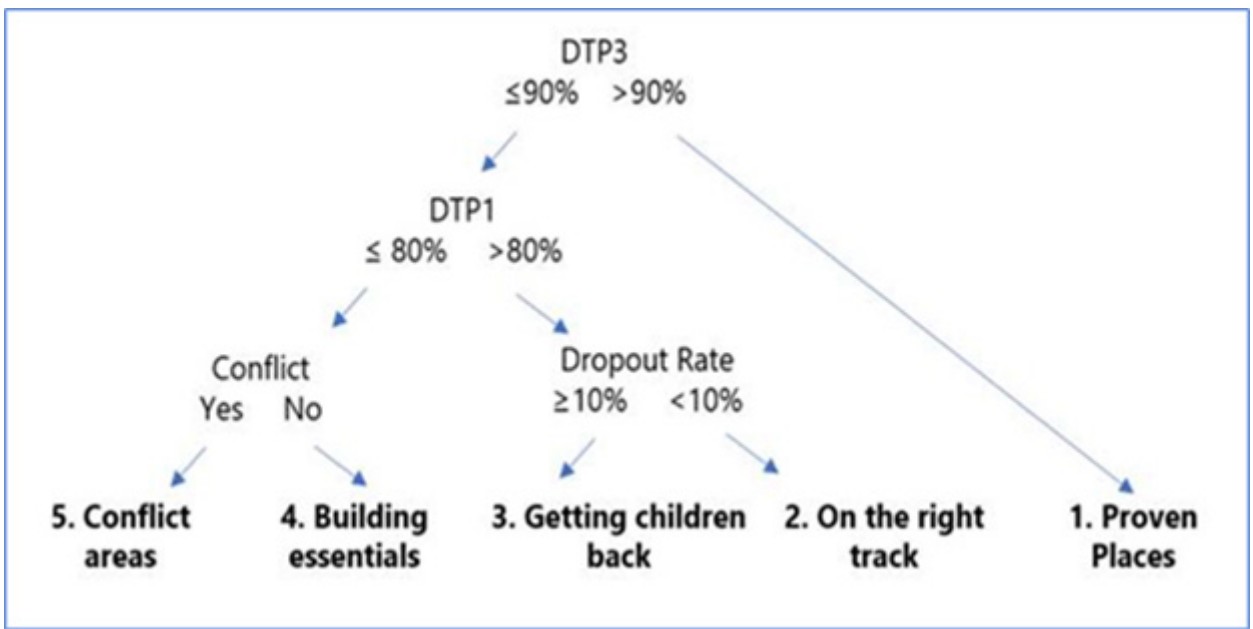

**Figure 2** Segment analysis logic. DTP, diphtheria, tetanus, pertussis.

### External advisory group

We formed a TAG consisting of experts in global health, vaccination delivery, vaccine confidence, and LIC and LMIC health systems to facilitate interpretation and dissemination of findings. The engaged stakeholder groups include WHO, UNICEF, CDC and Gavi. Engagement of the TAG is an ongoing process, with meetings convened for discussion at key decision points—including, but not limited to, input on final country selection, review of preliminary findings, review of context around key findings, and the current development of plans for dissemination

### Conceptual frameworks

This project uses several frameworks, which guided the development of tools and areas of inquiry. These overarching frameworks were taken from literature on vaccine delivery and implementation science. Implementation science is a growing field with the focus on applying evidence-based research findings into routine practice. Additional cross-cutting analyses use discipline-specific frameworks based on and extrapolated from the existing literature. The primary outputs of this study are country-level case studies, with additional cross-topic synthesis as possible.

### Vaccine delivery framework

Our conceptual model organises the complex interplay of barriers and factors impacting global childhood vaccine coverage, based on the work of Phillips *et al*[9] and LaFond *et al*,[10] and a broader review of the vaccine confidence and coverage literature (figure 4). Specific input was provided by our multidisciplinary team of public health, behavioural science, implementation science, political science, public policy and systems science and engineering researchers. This novel framework serves as

a guiding summary of the key issues for consideration in each country. The research is driven by the findings from each country (see Research Activities below), with no preconceptions regarding specific practices or interventions. An initial scoping visit for each exemplar country was used to gather preliminary feedback about the immunisation programme, historical challenges and interventions, and key stakeholders' initial impressions about reasons for success. These findings were then compared with the overall framework in figure 4 to identify specific areas in which additional focus was needed during the main research activities.

### Towards developing actionable recommendations

The goal of this project is to provide evidence-based, actionable recommendations to country and global stakeholders, with a focus on new insights to exemplary performance of vaccine delivery. Our initial scoping visits identified key historical barriers and interventions in each country; the focus of this research is understanding the 'how' and 'why' related to the adoption of each of these interventions or activities. Interventions may have been developed by stakeholders within each country (ie, endogenous innovation) or may be adaptations of higher-level guidance, such as local implementation of WHO guidance (ie, exogenous adaptation). For each intervention or programme—defined here as a solution developed and delivered by the country stakeholders ('what')—there is an iterative process between identifying the problem to be addressed ('why') and developing mechanisms for change, in other words 'how' the change could come about (figure 5). Understanding the interplay between 'how,' 'why' and 'what' can help identify actionable recommendations that may be useful for countries to consider when evaluating improvement in their vaccination systems.

**Table 1** Additional country selection criteria considered during study planning and rationale for final selection, as of 2018

| Region | Country | Inclusion decision | Rationale for inclusion decision | Selection method | Democracy index*[16] |
|---|---|---|---|---|---|
| Asia and South East Asia | India | No | Greater policy impact than Indonesia; unable to conduct research in-country | Both | Flawed democracy |
| | Indonesia | Potential Alternate | Less policy impact than India | CAGR | Flawed democracy |
| | Nepal | Yes | DTP3 gap closure and sustained high coverage | CAGR | Hybrid regimen |
| | Laos | Potential Alternate | Laos is an outlier in government type, so lessons will be less generalisable, signs of recent declines | Both | Authoritarian |
| East/Southern Africa | Zimbabwe | Potential Alternate | Possible systematic issues in coverage; Anglophone language group | Both | Authoritarian |
| | Burundi | No | Security concerns and access issues; Anglophone language group | Segment | Authoritarian |
| | Kenya | No | Higher trust in the data, more connections in country; Anglophone language group | Segment | Hybrid regimen |
| | Malawi | No | Small country, high coverage for a long period of time; Anglophone language group | Segment | Hybrid regimen |
| | Zambia | Yes | High DTP1 coverage maintained over the time period, closed gap between DTP1 and DTP3; Anglophone language group | Segment | Hybrid regimen |
| West Africa | Senegal† | Yes | Best option given difference in DTP3 and measles; relatively flat/downward since 2010, but signs of recent improvement; Francophone language group | Segment | Flawed democracy |
| | Burkina Faso | Potential Alternate | Relatively flat coverage—no change seen; Francophone language group | Both | Hybrid regimen |
| | Cameroon | No | Security concerns; Francophone language group | CAGR | Authoritarian |
| | Togo | Potential Alternate | Closing the gap between DTP1 and DTP3, but with slight declines in DTP1; Francophone language group | Both | Authoritarian |

*Terms from the Economist Democracy Index 2018, and briefly defined as follows: Flawed Democracies have free and fair elections, and basic civil liberties are respected even through problems and weaknesses in the system; hybrid regimens have elections with irregularities, contain weaknesses in the system and typically contain a weak civil society; Authoritarian Regimens do not have free and fair elections, if they occur at all, and infringe on civil liberties, along with repressing criticism and censoring dissenters.[16]
†As of the 2020 Democracy Index Report, Senegal is now considered a 'Hybrid Regimen'.[20]
CAGR, compound annual growth rate; DTP, diphtheria, tetanus, pertussis.

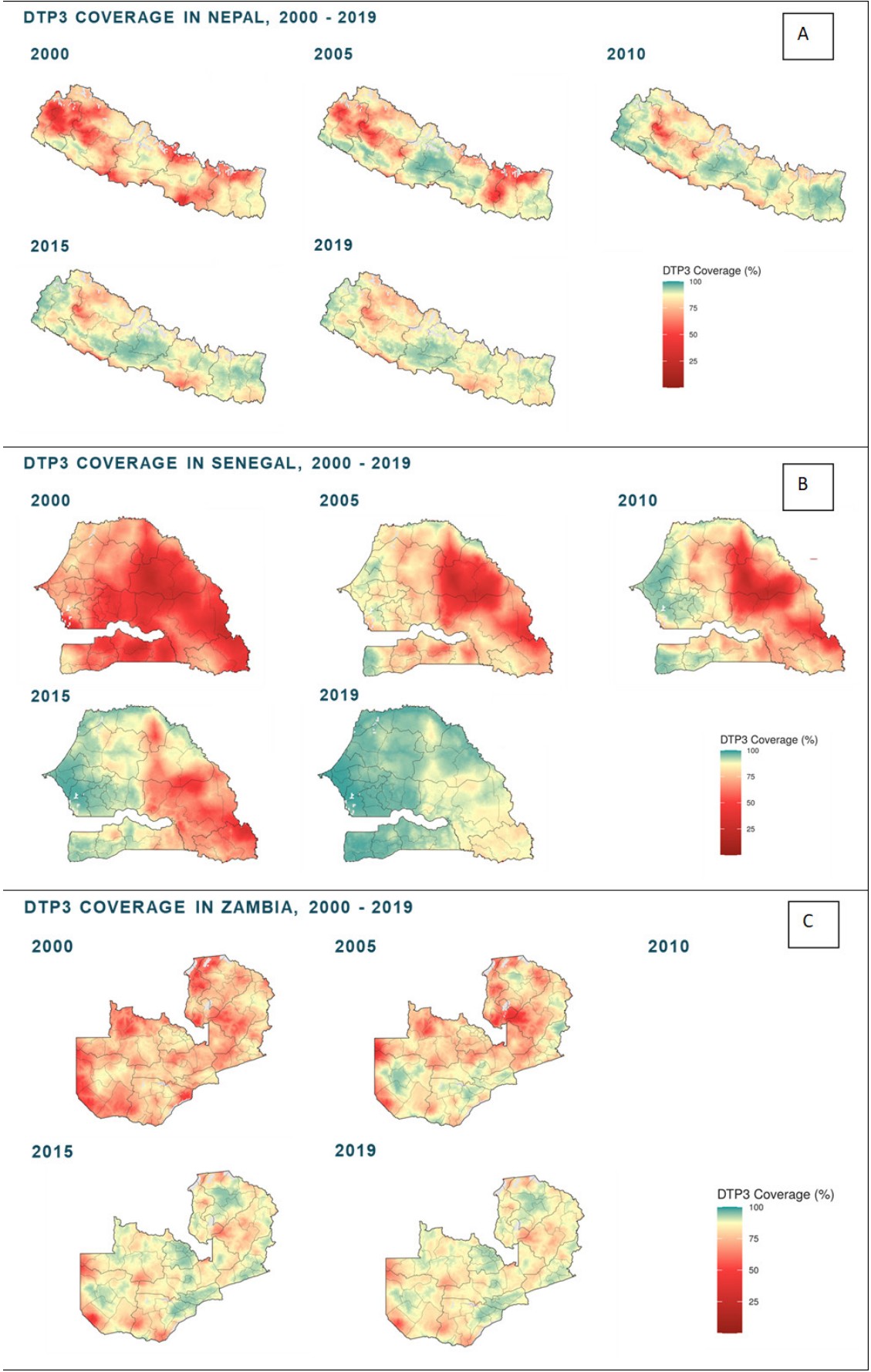

**Figure 3** Historical patterns of subnational DTP3 vaccine coverage in the three identified exemplar countries: Nepal (A), Senegal (B) and Zambia (C). DTP, diphtheria, tetanus, pertussis.

**Table 2** General summary of key informant and focus group participants by roles within the vaccine system

| | Nepal | Senegal | Zambia |
|---|---|---|---|
| **Interviews Key: no of KIIs (participants)** | 79 (79) | 63 (63) | 66 (85) |
| National level government staff | 11 (11) | 5 (5) | 11 (12) |
| Partner organisation staff | 8 (8) | 4 (4) | 11 (15) |
| Regional health staff | 14 (14) | 7 (7) | 6 (8) |
| District health staff | 15 (15) | 38 (38) | 10 (19) |
| Health facility staff | 10 (10) | 6 (6) | 7 (10) |
| Community leaders | 15 (15) | 2 (2) | 10 (10) |
| Community health workers* | 9 (9) | – | 11 (11) |
| **Focus groups Key: no of FGDs (participants)** | 30 (191) | 19 (128) | 22 (132) |
| Community health workers* | 9 (60) | 10 (65) | 10 (60) |
| Mothers | 9 (60) | 9 (63) | 8 (48) |
| Fathers | 6 (36) | – | 1 (6) |
| Grandparents | 6 (35) | – | 3 (18) |

*Includes volunteer community health workers, female community health volunteers, vaccinators, bajenu gox and neighbourhood health committee members.
FGD, focus group discussion; KIIs, key informant interview.

### Implementation science frameworks

A combination of two implementation science frameworks was applied to develop tools for data collection. Application of these frameworks directed our inquiry towards key domains of the historical decision-making and implementation process.

### Consolidated Framework for Implementation Research

Consolidated Framework for Implementation Research (CFIR) is a framework of five interrelated domains (intervention, outer setting, inner setting, individual characteristics and process of implementation) which influence the effectiveness of intervention implementation, and promote hypotheses of 'what works where and why across multiple contexts.'[17] We identified constructs within CFIR for focus within our tool development—including motivation, decision-making processes, mechanism for change, and the process and environment of development and delivery—in addition to inquires of events and policies most relevant to the success of Exemplar countries. The CFIR framework guides our examination of 'what they did,' 'why they did it' and 'how they did it,' at national,

regional, district and local levels in order to understand diverse contexts and perspectives within each of the exemplar countries. This allows us to systematically organise our findings, and better interpret the similarities and differences both across and between exemplar countries.

### Context and Implementation of Complex Interventions

The Context and Implementation of Complex Interventions (CICI) framework was applied in addition to CFIR to address contextual factors and the interdimensionality missing from the CFIR framework; both framed our initial thinking about the vaccine delivery system.[18] Both CFIR and CICI frameworks guided the development of an iterative data collection tool that could be applied consistently across diverse contexts and settings.

### Research activities

#### Tool development

Qualitative data collection was guided by semistructured key informant interview (KII) guides for use with health officials, external stakeholders and community leaders, and focus group discussion (FGD) guides for use with fathers, mothers, grandmothers and community health workers. These instruments explore the following CFIR and CICI domains: intervention characteristics, outer setting, inner setting, characteristics of individuals, process and context.[17] Qualitative data collection was intended to limit the time burden for KII or FGD participants to no longer than 1 hour, although some data collection took longer—up to 2 hours or more—based on the richness of the discussion. An initial KII guide was developed for scoping visits and was revised post visit to ensure data was captured within the domains of interest raised in those KIIs. Our overarching goal was to gather information from participants about 'how' and 'why' interventions were developed, adapted and implemented, and how they led to an increase in vaccination coverage. The guides were developed by the research team and refined through iterative review after completion of data collection in each country.

#### Scoping visits

Prior to beginning both in-depth data collection and review of relevant literature, we conducted a 2-week scoping visit in each country to (1) meet with and select in-country partners; (2) discuss key factors of change for further exploration (eg, identify the 'what' items for exploration of 'how' and 'why') and (3) prepare for in-depth country research activities (eg, establish local partnerships, start ethical reviews, research activity logistics).

#### Research visits and qualitative data collection

We conducted 10-day training workshops with our local research partners prior to the start of data collection in each country. In addition to training on study materials and methodology, we reviewed the materials alongside our in-country research partners to aid in any translation and adjust content for country context.

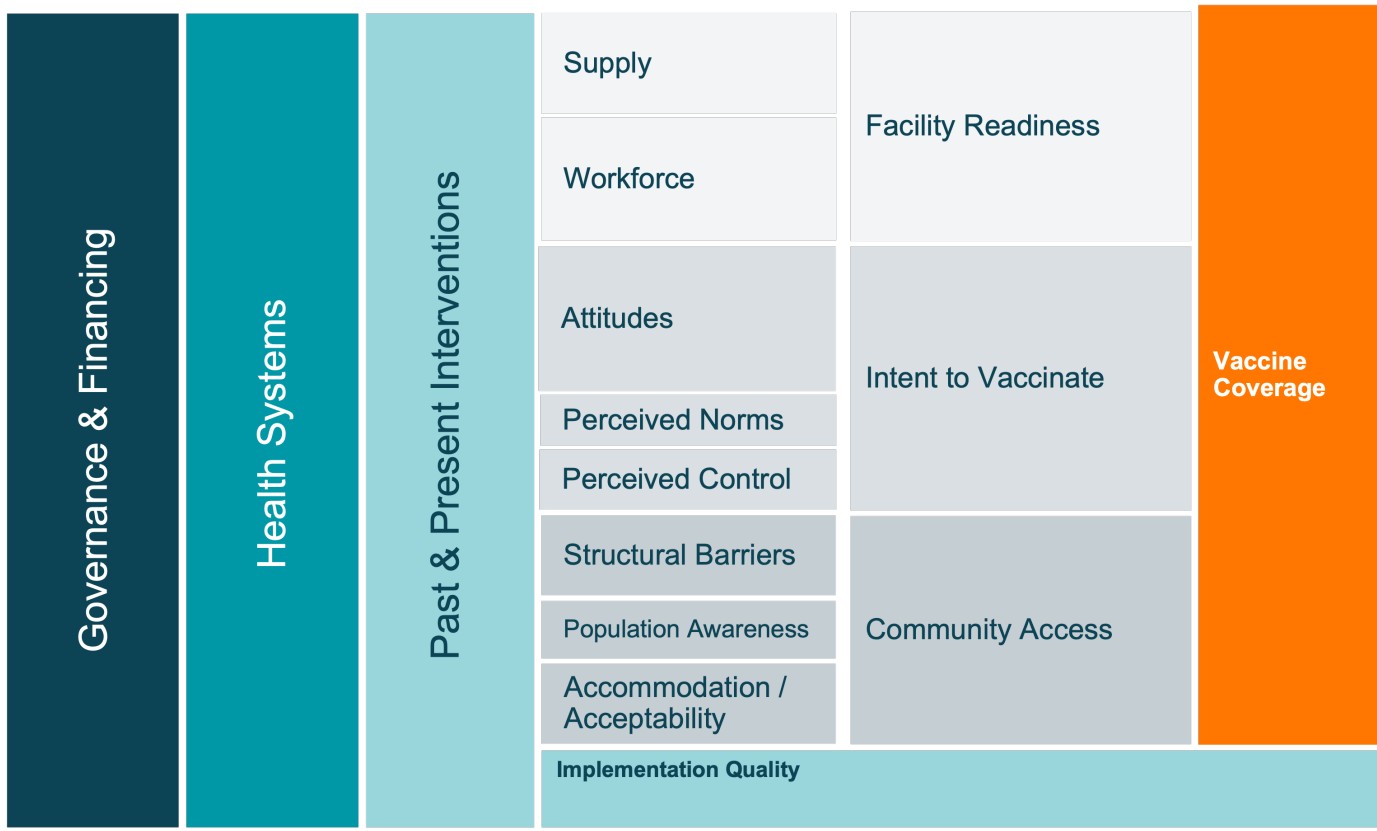

**Figure 4** Conceptual framework of drivers of vaccine delivery, derived from scoping visits, Phillips *et al*,[9] and LaFond *et al*.[10]

We conducted both KIIs and FDGs, as appropriate, with data collection occurring at the national level, subnational levels and community stakeholders at subnational levels (table 2). KIIs and FGDs took place in offices, clinics and community centres. All activities took place in a location deemed private, safe and comfortable by the participants. Qualitative data collection activities were conducted in person with trained facilitators and note-takers, when possible. Conditions for in-person research relative to the COVID-19 pandemic necessitated adjustments to maximise the quality of data collection and participant and researcher safety.

FGDs consisted of 6–8 participants. FGDs were held in the communities, organised by type of participant (eg, fathers will be in one group), and consisted of groups of fathers, mothers, grandmothers and community health workers. Partner organisations or community health workers identified the FGD participants.

### Qualitative data analysis and management
With permission from KII and FGD participants, interviews were recorded to ensure capture of all information. Recordings were transcribed verbatim from the local language by local research assistants and translated to English manually,

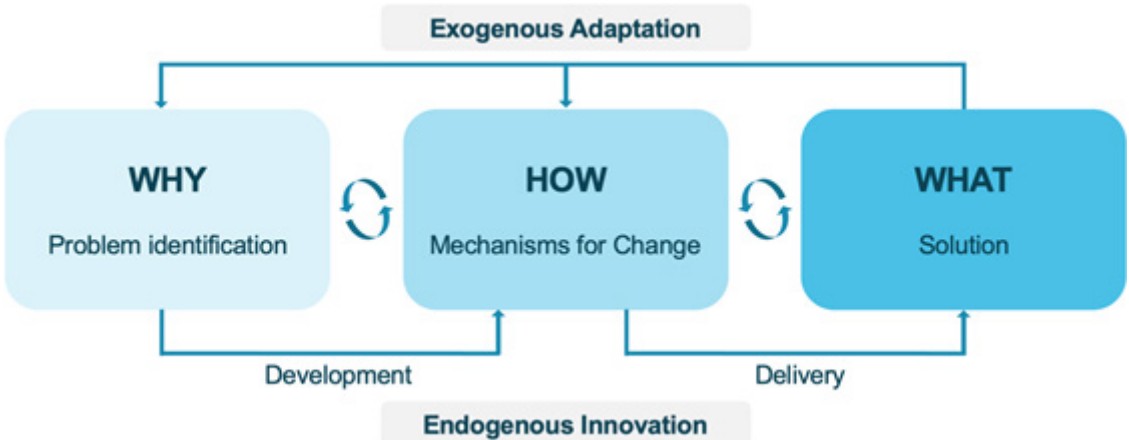

**Figure 5** Mapping the 'how' and 'why' behind an intervention.

or translated using Google Translate (for French), with verification by a fluent bilingual speaker. All documents with transcriptions were only accessible to researchers named on the IRB. All transcribed documents required a code to access. All research files, recordings and transcriptions in-country were saved on password-protected computers. Recordings were removed from recorders at the end of every day, deleted once uploaded onto password-protected computers and saved to HIPAA-compliant storage in folders only accessible to the study team. All recordings have been removed from computers and servers following transcription and verification of accuracy. Interviewees' names and contacts were deidentified, and all information will be used without mentioning their names. Documents that may link participants to their identifier code will be stored in separate locations.

Data were coded using MAXQDA V.20 (Berlin, Germany) and analysed thematically by specific aim, research question and framework-specific construct(s). The initial analysis for each country consisted of a case study, specific to that country, identifying the key drivers of improvements in vaccine coverage. This broad case study served as a starting point for more detailed topic-specific analyses and manuscripts. For key factors identified in multiple countries, a cross-country synthesis will be conducted to identify similarities and differences in implementation across study countries.

### Quantitative data collection

Quantitative data were gathered through freely obtained information on Ministry websites or data given from Ministry or other partners, such as the WHO, UNICEF and CDC. This quantitative analysis investigates vaccine coverage through a review of the health spending and economic growth trends from LICs and LMICs. Selected exemplar countries are compared with this grouping to determine what factors made exemplar countries stand apart from their peers. Analysis will use cross-country and multiyear mixed-effects regression models to statistically test financial, economic, development, demographic and other country-level indicators. A key component of this research will be to identify factors that may have been associated with improvements in vaccine coverage that are not commonly used as indicators of immunisation. This can include general health systems strengthening and improvements in funding for public health, as well as improvements in maternal and child health that may have driven support for immunisation services.[19]

### Patient and public involvement

We consulted with a TAG, but did not directly solicit patient or public involvement in the development of this research project.

## ETHICS AND DISSEMINATION
### Ethics

The study was approved by the Emory University Institutional Review Board (IRB); the Nepal Health Research Council in Nepal; the University of Zambia Biomedical Research Ethics Committee and the National Health Research Authority in Zambia; and the Comité de National d'Ethique pour la Recherche en Santé (CNERS; National Ethical Committee for Health Research) in Senegal. Participation in KII or FGD was voluntary, and interviewees were asked to provide informed consent.

### Dissemination

In addition to country-specific manuscripts describing our learnings, we will generate recommendations for national-level immunisation programmes based on the findings from this project. Specific reporting structures are listed below.

1. Country-level reports and case studies. The investigators will produce country-level findings, with feedback from country-level stakeholders. Country-level case studies will provide the basis for peer-reviewed manuscripts and broad dissemination on the Global Exemplars web platform.
2. Domain-level analysis. We will analyse each domain of interest identified from country case studies; these domains will be explored across exemplar countries. Current domains of interest for this synthesis include: targeted disease control activities, roles of community health workers and volunteers, health spending across LICs and LMICs, and intent and demand for vaccines. Findings will be disseminated among key national and global stakeholders and will be submitted for peer-review publication and for dissemination of the Gates Ventures web platform as cross-cutting synthesis.
3. Tool and protocol development. All individual frameworks and tools used by the research teams to inform research from their individual disciplines will be publicly available.
4. Knowledge translation and implementation outreach. Regional technical advisory meetings, webinars, policy fora, academic conferences, the exemplars platform and global partner meetings will be leveraged to disseminate findings. Additionally, findings will be translated into recommendations of replicable solutions for non-exemplar countries and areas for potential intervention investment for global immunisation actors and policy-makers. Documents might include policy briefs and infographics.
5. Exemplars in Global Health website. Exemplars.health is the platform documenting the work of the Exemplars in Global Health Project by Gates Ventures and will include narratives based on the research not just from the Vaccine Delivery project described here, but all other Exemplars in Global Health Projects.[12] The research team is working collaboratively with Gates Ventures to iteratively translate the research findings to the platform for public consumption.

## DISCUSSION

The Exemplars in Vaccine Delivery Project offers an opportunity to evaluate the critical factors in childhood vaccine delivery in LICs and LMICs. The in-depth qualitative

data collection and analysis will provide a deeper understanding of this issue based on the experiences and perspectives of key leaders in the three exemplar countries. Quantitative findings and existing literature will be used to triangulate findings. Our multi-disciplinary team brings experience in the fields of vaccine hesitancy, vaccine programme delivery, behavioural science, implementation science, public policy, political science, systems engineering. With a focus on changes over the previous two decades that may have spurred catalytic growth in vaccine coverage, these findings will present a unique opportunity to identify not just areas for improvement in global vaccine delivery, but the most appropriate methods to consider during implementation of these solutions. Longstanding efforts in health system strengthening offer a framework to build on, and the actionable recommendations that will arise from this project present a novel means to support the health of and protection from infectious diseases for children around the globe.

**Author affiliations**
[1]Hubert Department of Global Health, Rollins School of Public Health, Atlanta, Georgia, USA
[2]Emory Vaccine Center, Atlanta, Georgia, USA
[3]Gangarosa Department of Environmental Health, Emory University Rollins School of Public Health, Atlanta, Georgia, USA
[4]Center for Molecular Dynamics, Kathmandu, Nepal
[5]Department of Behavioral, Social, and Health Education Sciences, Rollins School of Public Health, Atlanta, Georgia, USA
[6]Zambia Emory HIV Research Project, Lusaka, Zambia
[7]Institut de Recherche et Santé de Épidémiologie et de Formations, Dakar, Senegal

**Collaborators** Vaccine Exemplars Research ConsortiumNatália S. Bueno, Bonheur Dounebaine, Kimberley R. Isett, Pinar Keskinocak, B. Pablo Montagnes, Dima Nazzal, Saad Omer, Walter Orenstein, Miguel R. Robayo, Simone Rosenblum, Francisco Castillo Zunino.

**Contributors** The study was conceived of by MCF and RAB; The protocol was developed by MCF, RAB, CE, KAH and ASE, Country selection was guided by KAH, MCF and RAB; country-level regional selection and tool adaptation was led by SMD, WK, MS, KM, KAH, ASE and ZMS; The first draft was written by RAB with editing by MCF, KAH, ASE, KM, ZMS, WK, SMD and MS. All authors provided input to facets of the overall study design and reviewed and approved the final manuscript.

**Funding** This work was supported by the Bill & Melinda Gates Foundation, grant number OPP1195041. Pilot and proposal development funds were provided by Gates Ventures.

**Map disclaimer** The inclusion of any map (including the depiction of any boundaries therein), or of any geographic or locational reference, does not imply the expression of any opinion whatsoever on the part of BMJ concerning the legal status of any country, territory, jurisdiction or area or of its authorities. Any such expression remains solely that of the relevant source and is not endorsed by BMJ. Maps are provided without any warranty of any kind, either express or implied.

**Competing interests** None declared.

**Patient and public involvement** Patients and/or the public were not involved in the design, or conduct, or reporting, or dissemination plans of this research.

**Patient consent for publication** Not applicable.

**Provenance and peer review** Not commissioned; externally peer reviewed.

**ORCID iDs**
Robert A Bednarczyk http://orcid.org/0000-0002-6812-0928
Moussa Sarr http://orcid.org/0000-0003-2372-6632

1  Greenwood B. The contribution of vaccination to global health: past, present and future. *Philos Trans R Soc Lond B Biol Sci* 2014;369:20130433.
2  Peck M, Gacic-Dobo M, Diallo MS, *et al*. Global routine vaccination coverage, 2018. *MMWR Morb Mortal Wkly Rep* 2019;68:937–42.
3  World Health Organization. Global Vaccine Action Plan 2011 - 2020 USA: World Health Organization; 2012.
4  World Health Organization. WHO-UNICEF estimates of DTP3 coverage; 2017.
5  WHO/UNICEF. Estimates of National Immunization Coverage; 2018.
6  Chandir S, Siddiqi DA, Mehmood M, *et al*. Impact of COVID-19 pandemic response on uptake of routine immunizations in Sindh, Pakistan: an analysis of provincial electronic immunization registry data. *Vaccine* 2020;38:7146–55.
7  Excler J-L, Privor-Dumm L, Kim JH. Supply and delivery of vaccines for global health. *Curr Opin Immunol* 2021;71:13–20.
8  Chiappini E, Parigi S, Galli L, *et al*. Impact that the COVID-19 pandemic on routine childhood vaccinations and challenges ahead: a narrative review. *Acta Paediatr* 2021;110:2529–35.
9  Phillips DE, Dieleman JL, Lim SS, *et al*. Determinants of effective vaccine coverage in low and middle-income countries: a systematic review and interpretive synthesis. *BMC Health Serv Res* 2017;17:681.
10  LaFond A, Kanagat N, Steinglass R, *et al*. Drivers of routine immunization coverage improvement in Africa: findings from district-level case studies. *Health Policy Plan* 2015;30:298–308.
11  Bradley EH, Curry LA, Ramanadhan S, *et al*. Research in action: using positive deviance to improve quality of health care. *Implement Sci* 2009;4:25.
12  Exemplars in Global Health, 2021. Available: https://www.exemplars.health/ [Accessed 26 May 2021].
13  Gavi. 2011-2015 indicators, 2020. Available: https://www.gavi.org/programmes-impact/our-impact/measuring-our-performance/2011-2015-indicators [Accessed 23 Mar 2022].
14  Shearer JC, Walker DG, Risko N, *et al*. The impact of new vaccine introduction on the coverage of existing vaccines: a cross-national, multivariable analysis. *Vaccine* 2012;30. doi:10.1016/j.vaccine.2012.10.036. [Epub ahead of print: Available from] https://www.who.int/immunization/sage/meetings/2012/april/6_ExecSumDTP3_coverage.pdf
15  Institute for Health Metrics and Evaluation. IHME national coverage estimates unpublished; 2019.
16  The Economist Intelligence Unit. Democracy index 2017: free speech under attack; 2018.
17  Damschroder LJ, Aron DC, Keith RE, *et al*. Fostering implementation of health services research findings into practice: a consolidated framework for advancing implementation science. *Implement Sci* 2009;4:50.
18  Pfadenhauer LM, Gerhardus A, Mozygemba K, *et al*. Making sense of complexity in context and implementation: the context and implementation of complex interventions (CICI) framework. *Implement Sci* 2017;12:21.
19  Castillo-Zunino F, Keskinocak P, Nazzal D, *et al*. Health spending and vaccination coverage in low-income countries. *BMJ Glob Health* 2021;6:e004823.
20  The Economist Intelligence Unit. Democracy index 2020: in sickness and in health? 2021.

