## [Reviewer comments · BMJ Open]

ARTICLE DETAILS

TITLE (PROVISIONAL)	Exemplars in Vaccine Delivery Protocol: A Case-Study-based Identification and evaluation of critical factors in achieving high and sustained childhood immunization coverage in selected low- and lower-middle income countries
AUTHORS	Bednarczyk, Robert A.; Hester, Kyra; Dixit, Sameer; Ellis, Anna; Escoffery, Cam; Kilembe, William; Micek, Katie; Sakas, Zoë; Sarr, Moussa; Freeman, Matthew

VERSION 1 – REVIEW

REVIEWER	Eliana Biundo GSK Belgium
REVIEW RETURNED	06-Jan-2022

GENERAL COMMENTS	line 117 - it is unclear what the 3 areas of inquiry are. the three elements described are all related to critical policy and programmatic innovations. Please clarify. lines 131-133 - it might be good to add a reference to other studies using DTP coverage as proxy of vaccination program performance. line 136 - reference to CAGR calculation based on 3 year average while in line 130 it mentions 5 years. line 153 - the process on how the final 3 countries are selected is unclear. Please add details on how the criteria used for the selection and if there was expert input (the beginning of the methods section seems to suggest that but here it is not mentioned). Figure 4 - unclear why in the framework there are mentioned dropouts, left-outs and missed opportunities. it is a generic framework that depending on data from a given country might explain what lead to dropouts but this should not be part of the framework per-se in my opinion. also in Line 189 you mention it is based on the work of Phillips et al. and LaFond et a. but in the caption to Figure 4 you mention only the former reference. Please be consistent. Unclear how many KIIs and FDGs were conducted. It remains unclear if future data collection is planned or if the protocol described the data collection activities already performed and what outcomes the study will produce. if data has already been collected it might be useful to describe more in detail how these will be summarised and analysed to build the study results. Suggestion to reflect also on what has been the impact of the COVID-19 pandemic and if any of the identified best-practices has been successful also during the pandemic. This might enable in the recommendations to point out which of the multiple best-practices is 'stronger'.
---

REVIEWER	Angela Oyo-Ita University of Calabar, Nigeria
REVIEW RETURNED	29-Jan-2022

GENERAL COMMENTS	Authors addressed a topical issue that has the potential of impacting on policy and practice as it relates to improving vaccine coverage. Rigorous search strategy was applied. Available evidence were well mapped and the gaps identified. Funding should be included in the theory of change (Fig 1) under health system oriented interventions.
---

VERSION 1 – AUTHOR RESPONSE

Reviewer # 1		
Comment	Response	Revision location
1. line 117 - it is unclear what the 3 areas of inquiry are; the three elements described are all related to critical policy and programmatic innovations. Please clarify.	We have modified the sentence for clarity Modified text: This study focuses on critical policy and programmatic innovations that drove changes to vaccine coverage and equity across the three countries of interest, and specifically investigates “how” and “why” these innovations were implemented.	Page 4, lines 118-121
2. lines 131-133 - it might be good to add a reference to other studies using DTP coverage as proxy of vaccination program performance.	We have added citations on guidelines/definitions of use of DTP1/DTP3 from Gavi and WHO	Page 4, line 135
3. line 136 - reference to CAGR calculation based on 3-year average while in line 130 it mentions 5 years.	We removed the phrase “over 5 year increments”, this was an artifact of our initial data review.	Page 4, lines 131-132
4. line 153 - the process on how the final 3 countries are selected is unclear. Please add details on how the criteria used for the selection and if there was expert input (the beginning of the methods section seems to suggest that but here it is not mentioned).	Table 1 goes into more detail regarding our selection process. We have added a sentence stating country selection was aided by our technical advisory group. Added text: Final exemplar countries were selected in conjunction with our Technical Advisory Group.	Page 4, lines 158-159.

Reviewer # 1		
Comment	Response	Revision location
5. Figure 4 - unclear why in the framework there are mentioned dropouts, left-outs and missed opportunities. it is a generic framework that depending on data from a given country might explain what lead to dropouts but this should not be part of the framework per-se in my opinion. also in Line 189 you mention it is based on the work of Phillips et al. and LaFond et a. but in the caption to Figure 4 you mention only the former reference. Please be consistent.	We have updated our framework without that detail, and have added reference to LaFond et al. to the caption	Figure 4 Figure 4 caption
6. Unclear how many KIIs and FDGs were conducted. It remains unclear if future data collection is planned or if the protocol described the data collection activities already performed and what outcomes the study will produce. if data has already been collected it might be useful to describe more in detail how these will be summarised and analysed to build the study results.	Table 2 is updated to reflect all counts of KII and FGD. As of the time of this resubmission, all data collection activities have been completed.	Table 2
7. Suggestion to reflect also on what has been the impact of the COVID-19 pandemic and if any of the identified best-practices has been successful also during the pandemic. This might enable in the recommendations to point out which of the multiple best-practices is 'stronger'.	This study was conceptualized, and data collection started, prior to the COVID-19 pandemic. Only one of the countries, Senegal, had data collection affected by the pandemic – this is addressed in the individual case study on this country that is currently under review at Vaccine X. We feel that we cannot accurately speak to the impact of COVID due to timing and the retrospective study design, which explores historical interventions with a time period of close to 20 years (2000 – 2018). We explored the effects of COVID in as separate paper, which can be found here	N/A

Reviewer # 2		
Comment	Response	Revision location
1. Authors addressed a topical issue that has the potential of impacting on policy and practice as it relates to improving vaccine coverage. Rigorous search strategy was applied. Available evidence were well mapped and the gaps identified. Funding should be included in the theory of change (Fig 1) under health system oriented interventions.	We thank the reviewer for their positive comments. Funding is included, conceptually, under the “Governance and Financing” portion of our framework.	N/A

VERSION 2 – REVIEW

REVIEWER	Eliana Biundo GSK Belgium
REVIEW RETURNED	30-Mar-2022
GENERAL COMMENTS	No further comments. Thank you for having incorporated my suggestions in this draft.